# Exploring Rare Disease Patient Attitudes and Beliefs regarding Genetic Testing: Implications for Person-Centered Care

**DOI:** 10.3390/jpm12030477

**Published:** 2022-03-16

**Authors:** Andrew A. Dwyer, Melissa K. Uveges, Samantha Dockray, Neil Smith

**Affiliations:** 1Massachusetts General Hospital—Harvard Center for Reproductive Medicine, Boston, MA 02114, USA; 2William F. Connell School of Nursing, Boston College, Chestnut Hill, MA 02467, USA; uveges@bc.edu; 3School of Applied Psychology, University College Cork, T12 YN60 Cork, Ireland; s.dockray@ucc.ie; 4HYPOHH Patient Support Group, London WD3 1FX, UK; neilsmith38@hotmail.com

**Keywords:** decision making, ethics, genetic counseling, genetic testing, hypogonadotropic hypogonadism, Kallmann syndrome, person-centered care, qualitative research methods, rare disease, risk communication

## Abstract

Most rare diseases are genetic in etiology and characterized by a ‘diagnostic odyssey’. Genomic advances have helped speed up the diagnosis for many rare disorders, opening new avenues for precision therapies. Little is known about patient attitudes, experiences, and beliefs about genetic testing for the rare disease congenital hypogonadotropic hypogonadism (CHH). Methods: We conducted six focus groups with patients with CHH (*n* = 58). Transcripts were coded by independent investigators and validated by external reviewers. Results: Major themes relating to pre-test experiences were ‘attitudes & beliefs’ (most frequently cited theme), which revealed altruism as a strong motivator for pursuing research testing and ‘information and support,’ which revealed a striking lack of pre-testing decisional support/genetic counseling. Major post-test themes included ‘return of results,’ revealing frustration with the lack of return of results and limited emotional support, and ‘family communication,’ describing challenging intrafamilial communication. Themes describing ethical concerns (i.e., privacy, use of samples) were least frequently noted and related to pre- and post-test experiences. Conclusions: Patients with CHH are highly motivated by altruism when pursuing testing but have significant unmet needs for pre-test decisional support and post-test counseling. It is regarded that patient values, beliefs and experiences can inform more person-centered approaches to genetic testing for rare diseases.

## 1. Introduction

Rare diseases are often characterized by a “diagnostic odyssey” [1]. From initial presentation, the average time to arrive at a correct rare disease diagnosis is five years [2]. Such diagnostic delays can cause significant psychological, emotional, and financial distress as well as delays in initiating care and treatment. As most rare diseases are genetic in etiology, the growth of next-generation sequencing offers new avenues to accelerate diagnosis for rare diseases [3,4].

Congenital hypogonadotropic hypogonadism (CHH) is a rare endocrine disorder that is clinically characterized by absent/incomplete puberty and infertility resulting from deficient secretion (or action) of gonadotropin-releasing hormone (GnRH) [5]. A range of non-reproductive phenotypes are associated with CHH. Approximately half of patients exhibit a significantly impaired or absent sense of smell (anosmia) [6]. When CHH occurs with anosmia, it is termed Kallmann syndrome (KS). A range of other non-reproductive phenotypes have been reported in the literature at variable rates including midline defects (cleft lip/palate), skeletal/dental anomalies, unilateral renal agenesis, synkinesia (mirror movement), and hearing loss [5]. Current estimates suggest that CHH occurs in approximately 1 in 48,000 people [7] with a 4:1 (male:female) sex discordance [8].

Unlike many rare diseases, effective treatments are available for inducing secondary sex characteristics (i.e., sex steroid replacement), and fertility can be induced in roughly 75–80% of men with either gonadotropin injections or pulsatile GnRH [9]. Similar success rates are observed in women with CHH/KS with gonadotropin therapy or pulsatile GnRH [5]. However, like other rare diseases, delayed diagnosis is common. In CHH/KS, later diagnosis is associated with increased psychosocial morbidity and psychosexual concerns [10,11]. Delays in diagnosis are partly due to the fact that CHH/KS is a diagnosis of exclusion and there is no “gold standard” test to differentiate delayed puberty from CHH/KS [12]. In some cases, “red flags” such as anosmia or absent ‘minipuberty’ in males during the first 6 months of life (as evidenced by cryptorchidism with/without micropenis) can cue a knowledgeable clinician towards a CHH/KS diagnosis [13]. Unfortunately, all too often, a “watchful waiting” approach is taken, delaying both diagnosis and timely initiation of treatment—resulting in significant patient distress and impaired health-related quality of life [10,11,14].

Genetic testing has the potential to surmount some of the diagnostic challenges and hasten detection, thereby helping to alleviate the patient’s psychosocial burden [4]. In the absence of the aforementioned clinical “red flags”, identifying a rare variant in a CHH/KS loci could support a probable diagnosis, initiation of treatment, and help ease the patient’s psychosocial distress. Similar to the clinical variability of CHH/KS, the condition is also genetically heterogeneous. To date, more than 60 loci have been identified to underlie CHH/KS, accounting for approximately half of cases [15]. Patients may seek genetic testing through ongoing research programs as well as through commercial diagnostic laboratories. From the initial genetic discovery of identifying CHH/KS as a monogenic X-linked condition, the genetics of CHH/KS have grown increasingly complex. In addition to X-linked inheritance, autosomal dominant and recessive inheritance have been reported as well as an increasingly complex genetic architecture including digenic and oligogenic forms [15]. As such, few clinicians outside specialist centers have a deep understanding of the genetics of CHH/KS, which limits patient knowledge and understanding. While genetic discovery over the past decades has deepened our understanding of the molecular basis of CHH/KS, little is known about patient and family experiences and perspectives regarding genetic testing for CHH/KS.

In this study, we sought to explore patient experiences, attitudes, and beliefs regarding genetic testing for CHH/KS as well as communication with potentially at-risk blood relatives. A deeper understanding of patient perspectives on genetic testing could inform more person-centered approaches to supporting patients in navigating genetic testing services and decision-making.

## 2. Materials and Methods

This prospective qualitative study involved both face-to-face (*n* = 3 in-person, 33 total participants) and online (*n* = 3 virtual, 25 total participants) focus group discussions. The study was conducted in accordance with the Declaration of Helsinki. The study protocol was reviewed and approved by the ethics committee of the University of Lausanne (protocol #2016_02184) and the Boston College Institutional Review Board (protocol #18.081.01). All participants provided informed consent prior to study participation.

### 2.1. Participants

We employed a community-based participatory research framework [14] for the purposive sample of focus group participants including patients (with CHH/KS) and parents/guardians of children/adolescents with CHH/KS. Briefly, we partnered with a CHH/KS patient community leader to co-organize and conduct informational patient meetings (in-person and virtual). Patient meetings provide general information about CHH/KS including an overview of pathophysiology, genetics, diagnosis, treatment options, living with CHH/KS, and health promotion topics. At the close of the informational meeting, English-speaking adult patients (18 years and older) and parents/guardians of a child/adolescent with CHH/KS were invited to participate in a focus group discussion. Prior to study participation, participants provided written (for in-person focus groups) or opt-in electronic (for virtual focus groups) consent, respectively. Participants were offered a USD 25 remuneration for study participation.

### 2.2. Focus Group Discussions

Semi-structured focus group discussions (90–120 min) were led by an investigator (AAD) and the patient group leader (NS). Question prompts (Appendix A) related to experiences around genetic testing for CHH/KS, issues related to decision-making, and discussing CHH/KS with potentially at-risk blood relatives. The in-person focus groups were conducted during annual CHH/KS patient meetings (2017–2019). Virtual focus groups were conducted over Zoom (January to June 2021). All focus group discussions were audio recorded and transcribed verbatim. Patient participants in the transcripts were de-identified (i.e., assigned participant numbers). A concept for qualitative inquiry (i.e., focus groups) is saturation—meaning the time at which no new codes emerge. Typically, three to five focus group discussions are needed to reach saturation [16]. Thus, we conducted six focus groups to ensure data saturation.

### 2.3. Analysis

We employed a coding reliability approach to thematic analysis [17]. To create the codebook for analyzing focus group transcripts, independent investigators (AAD, MU) reviewed and coded the first transcript using Dedoose (Version 9.0.17, SocioCultural Research Consultants LLC, Los Angeles, CA, USA, www.dedoose.com (accessed on 17 February 2022)). Briefly, investigators highlighted excerpts (sections of meaningful content in the transcript) and labeled excerpts with an identifying code (representing a theme or sub-theme). Subsequently, investigators met and created the codebook by discussing a hierarchy of themes/sub-themes, collapsing similar codes, and operationalizing theme/sub-theme definitions. Investigators then used the codebook to independently code the remaining transcripts. To ensure rigor, we used external reviewers to “triangulate” and validate the independent coding results. The patient group leader (NS) and an investigator experienced in qualitative research (SD) served as external reviewers who evaluated transcripts and the list of themes/sub-themes to determine if all concepts were appropriately captured.

Participant characteristics are reported using descriptive statistics. Comparisons between in-person and virtual groups were performed using Chi-square/Fisher’s Exact and Student’s *T*-test/Mann–Whitney U test as appropriate. A *p* value < 0.05 was considered statistically significant. Theme/sub-theme frequencies were calculated to determine patient-identified priority areas.

## 3. Results

Six focus group discussions were conducted including three in-person and three virtual. Focus group participant characteristics are reported in Table 1. Focus group codes were organized into themes (reflecting major concepts), sub-themes (reflecting theme components), and dimensions (reflecting sub-theme elements) (Figure 1). Findings on genetic testing for CHH/KS are reported in three broad categories: pre-test experiences, post-test experiences, and ethical concerns.

### 3.1. Pre-Test Experiences

Two dominant themes emerged related to pre-test experiences. ‘Attitudes & beliefs’ relates to motivating factors (for/against) genetic testing). This theme includes four sub-themes: motivating factors (for testing), test type (diagnostic vs. research), uncertainty (i.e., unanticipated/incidental findings), and cost of genetic testing. ‘Information and support’ relates to where patients learned of genetic testing and experiences with decisional support for genetic testing. This theme includes three sub-themes: information source, pre-test decisional support, and genetic counseling (Figure 1).

#### 3.1.1. Theme: ‘Attitudes & Beliefs’

The ‘attitudes & beliefs’ theme was the most frequently cited in focus group discussions (Figure 1). The sub-theme ‘motivating factors’ appeared 204 times. ‘Motivating factors’ to pursue genetic testing include gaining a deeper understanding of the condition and clearing up diagnostic uncertainty (i.e., helping to end the “diagnostic odyssey” and “putting a name” on the problem). Another ‘motivating factor’ for testing was altruism (Table 2). Some patients sought to understand their genetics to inform and help younger, potentially at-risk blood relatives (e.g., children, nieces, nephews). Altruism was also a strong motivator for participating in genetic research to help investigators/clinicians better understand the molecular basis of CHH/KS. Given the psychological impact of the condition on their lives, participants deeply hoped that contributing to scientific knowledge would benefit future generations of patients (Table 2). Factors dissuading participants from opting for genetic testing included perceptions that genetic testing would not inform/guide treatment or that testing would not improve their overall wellbeing and quality of life (Table 2). The sub-theme ‘uncertainty’ (i.e., incidental or unanticipated findings) was another aspect that factored into participants not pursuing genetic testing (*n* = 30 mentions). Relating to the sub-theme ‘test type’ (*n* = 50 mentions), approximately half of participants (27/58, 46%) had undergone genetic testing for CHH/KS. Of those who had testing, the overwhelming majority (25/27, 93%) did so in a research setting. Only two participants had testing in the context of a clinical diagnostic lab. The rarity of clinical diagnostic testing among participants likely relates to the sub-theme ‘cost’ (*n* = 21 mentions). Indeed, a number of patients reported cost as the single biggest barrier preventing them from receiving genetic testing and expressed concerns that insurance would not cover genetic testing. The strong role that altruism plays as a motivator for pursuing genetic testing (particular in the research context) is a notable finding emerging from the focus group discussions.

#### 3.1.2. Theme: ‘Information and Support’

Under the theme ‘information and support,’ the sub-theme ‘information source’ appeared 82 times across the six focus groups. Focus group participants reported learning about genetic testing for CHH/KS from two primary sources—their specialist (endocrinologist) or via online discussions in the patient support group’s private Facebook page. In total, 58 participants had undergone genetic testing. A number of participants expressed frustration that they were not offered genetic testing (Table 3). Similarly, two additional sub-themes describe participants’ disappointment and frustration with the paucity of pre-test decisional support (*n* = 40 mentions) and lack of pre-test genetic counseling (*n* = 30 mentions). The dearth of pre-test decisional support is highlighted by the observation that only 4/58 (7%) participants had met with a geneticist or genetic counselor prior to genetic testing. No participants reported receiving virtual/online decisional support. Participants also expressed a desire for counseling and support to help them make better-informed decisions about genetic testing (Table 3). The lack of pre-test decisional support is a notable unmet patient need emerging from the focus group discussions.

### 3.2. Post-Test Experiences

Two dominant themes related to post-test experiences. First, ‘return of results’ includes four sub-themes: uncertainty about test results, difficulty understanding results, waiting for results, and lack of emotional support following genetic testing. Second, ‘family communication’ includes the sub-themes of perceived promoters and barriers to discussing CHH/KS with family members (i.e., potentially at-risk blood relatives).

#### 3.2.1. Theme: ‘Return of Results’

The theme ‘return of results’ was frequently cited in focus group discussions. The sub-theme ‘uncertainty’ (*n* = 85 mentions) relates to patient concerns about having a negative test result that would not fully elucidate the cause of their condition (Table 4). Participants expressed concern that even though they had been diagnosed with CHH/KS, their genetic test result could be negative (i.e., no rare variants in CHH/KS loci). Similarly, some participants expressed that they did not know what to do with the results in relation to healthcare providers and family (Table 4). The sub-theme ‘results interpretation’ (*n* = 67 mentions) relates to patients feeling overwhelmed by the complexity of the genetic test results and feeling unsupported and “on their own” in understanding the implications of genetic test results (Table 4). A related sub-theme was ‘lack of post-test support’ (*n* = 37, mentions). Participants expressed feelings of disappointment and sadness associated with a perceived lack of emotional support following genetic testing (i.e., traditional post-test genetic counseling). Several participants also expressed a desire for coaching and counseling to help cope with living with CHH/KS as well and how to discuss the diagnosis with potentially at-risk blood relatives (Table 4). Participants were particularly impassioned about the sub-theme ‘lack of results/waiting’ (*n* = 54 mentions). Participants expressed frustration and disappointment that they had “done their part” (i.e., providing DNA)—yet they rarely (if ever) received results (Table 4). Extended waiting to hear back from researchers and having to “badger” investigators to obtain information were a common and shared source of discontent. Patient frustration with waiting for results and the lack of information regarding test results is a key finding emerging from the focus group discussions.

#### 3.2.2. Theme: ‘Family Communication’

Focus group discussion revealed both facilitators (promoters) and barriers to intrafamilial communication regarding CHH/KS diagnosis. Barriers were more frequently discussed than promoters of family communication (*n* = 109 vs. *n* = 65 mentions respectively). Male patients, female patients and parent/guardians alike shared stories highlighting the shame related to a rare disease diagnosis. Unaccepting family members (i.e., denial) and personal feelings of shame were often cited as barriers to intrafamilial communication about CHH/KS (Table 5). Many patients shared intimate stories of how family members refused to accept that a member of the family had CHH/KS, considering the diagnosis a “deep, dark secret” not to be discussed. A number of patients were emotionally traumatized by seeing peers develop while they were essentially “trapped” in a child’s body. For many, the impact of psychosocial trauma persisted despite having long-term sex steroid treatment—that effectively induces secondary sex characteristics but not fertility. Patient-identified factors supporting intrafamilial communication included family dynamics (i.e., supportive of each other, open communication patterns), encouragement and support from other patients (in the form of online peer-to-peer support), and a strong sense of altruism to potentially help younger family members (i.e., informing family could help earlier detection in affected nieces/nephews). Notably, several participants offered suggestions for resources and tools (e.g., printed information, brief videos, online consultation with an expert in CHH/KS) to help support intrafamilial communication about CHH/KS (Table 5). Specific facilitators/barriers to intrafamilial communication as well as the lack of support/guidance for having discussions around CHH/KS were key findings emerging from the focus group discussions.

### 3.3. Theme: ‘Ethical Concerns’

The least frequently cited theme related to ‘ethical concerns.’ All but one sub-theme (informed consent) related to post-test ethical concerns. The sub-theme ‘privacy and data use’ arose most often (*n* = 44 mentions). Participants expressed significant concerns about threats to their privacy and confidentiality (i.e., their identity could be exposed). A number of participants claimed distrust of for-profit organizations (i.e., industry, health insurers) and worried that their information could be used in harmful ways (i.e., increased insurance premiums) (Table 6). Given the rarity of CHH/KS, samples are sometimes shared between laboratories for research. Some participants raised the sub-theme of ‘sample traceability’ (*n* = 24 mentions) relating to concerns that their DNA would be shared and passed between laboratories without patient consent/permission (Table 6). Several participants also expressed a desire for a greater role in decision-making regarding research (i.e., having a voice in determining what tests were administered, if they would receive results, and who could have access to patient specimens). The sub-theme ‘informed consent’ (*n* = 20 mentions) relates to pre-test ethical concerns. Participants noted that consent was treated as a document to be signed rather than a process (Table 6). Concerns around privacy and confidentiality as well as a lack of a coherent process for informed consent are notable unmet patient needs emerging from the focus group discussions.

## 4. Discussion

Herein, we report patient and parent/guardian experiences, attitudes, and beliefs regarding genetic testing for a rare endocrine disorder (CHH/KS). We observed similar rates of genetic testing in our participants compared to prior prospectively recruited patients with CHH/KS (43/100, 43%, *p* = 0.79) [18]. Of participants who had undergone genetic testing in the present study, the majority (93%) had done so in a research context. The cost of clinical diagnostic genetic testing was identified as a barrier to testing. A key finding related to pre-test experiences was the notable paucity of pre-test decisional support/genetic counseling. Genetic counseling is an important aspect of supporting high-quality genetic testing decisions—i.e., decisions that are both informed and aligned with the patient values and preferences. Genetic counseling typically involves both pre- and post-test consultations. Pre-test counseling provides non-directive techniques to educate patients about potential risks/benefits, possible test results and their implications, as well as the limitations of genetic testing [19]. As such, genetic counselors seek to provide clear, understandable information, elicit client values/beliefs, and invite reflection to support high-quality decisions. Post-test consultations serve to debrief clients to help them understand their test results and what to do with the findings (i.e., how results may affect treatment and reproductive decisions). However, traditional genetic counseling is not always part of genetic research studies.

In the present study, patients/families expressed frustration with the lack of services—pointing to an unmet need for decisional support. Genetic counselors are typically well-versed in common genetic conditions—i.e., tier 1 conditions including hereditary breast and ovarian cancer (HBOC), Lynch syndrome and familial hypercholesterolemia [20]. However, there are an estimated 7000 rare diseases and finding counselors with expertise in a specific rare disease such as CHH/KS can be difficult. Indeed, only 4/58 (7%) participants had met with a geneticist or genetic counselor prior to genetic testing—consistent with rates observed in previous studies of patients with CHH/KS (13/101 13%, *p* = 0.36) [18]. The genetic complexity of CHH/KS (i.e., variable expressivity, incomplete penetrance, oligogenicity) poses additional challenges for genetic counselors [15]. Indeed, there are only two published articles on genetic counseling for CHH/KS [21,22]. Currently, there is a shortfall of trained healthcare professionals to meet the burgeoning demand for genetic counseling [23]. Only a handful of countries meet the United Kingdom Royal College of Physicians recommendation of 6–12 genetic counselors per million persons [24]. In response, there is a growing interest in ‘telegenetics’ [25] and the use of online decision-support tools [26]. As rare-disease patients have been termed internet “power-users” [27], online decisional support is a possible solution responding to the shortage of genetic counselors, geographic distance, and the specialized nature of CHH/KS genetic counseling. In addition to providing information and values reflection, online support could offer behavioral theory-informed coaching for starting discussions with family—as has been used in HBOC [28]. Such a tool aligns with the desires of patients (Table 3, Dimension: desired supports for communication (tools/resources)) and could also help address other sub-themes emerging from focus group discussions including ‘uncertainty’ and ‘barriers to family communication’.

Another key finding regarding pre-test experiences relates to altruism as a strong motivator for pursuing genetic testing (particularly in the research context). The psychosocial impact of CHH/KS has been well described [8,10,11,14,29]. Patients expressed a strong desire to do whatever they could to alleviate the burden of CHH/KS from future generations. The dimension of ‘altruism’ was particularly notable among older patients who were focused on sparing other patients the stigmatizing and sometimes traumatic experiences that many of the older patients had experienced. Altruism was also a factor that motivated family discussions. These observations are similar to qualitative findings in families with HBOC [30,31]—wherein some individuals feel obligated to communicate risk within the family and become so-called “*BRCA* warriors”, informing at-risk blood relatives. Importantly, feelings of guilt often accompany a genetic diagnosis [29,32]. The guilt and shame noted by participants mirrors sentiments expressed in HBOC families and poses barriers for intrafamilial communication [29,30,31,33]. Focus-group discussions revealed conflicting feelings as many were altruistically motivated to participate in genetic research and help investigators/clinicians—yet participants were disappointed by some providers being unaware/not offering genetic testing. Indeed, genetic testing may be important for informing reproductive decisions for those patients who are considering fertility-inducing treatment. Indeed, translational roadblocks contributing to the seventeen-year lag from discovery to implementation into practice are well known [34]. Recently, implementation frameworks have been developed to better integrate genetic services into primary care [35]. Such efforts may help ameliorate some of the referral barriers expressed by focus group participants.

In relation to post-test experiences, the extended wait for the return of results and lack of communication/results was a major finding and a significant source of frustration (Table 4). Interestingly, parents/guardians expressed a particular emphasis on post-test experiences—primarily, the theme ‘family communication’ and concern for their child post-test (i.e., the dimension ‘lack of understanding about the impact of diagnosis’). A 2020 study explored facilitators/barriers to genetic research participation among more than 57,000 members of the Kaiser Permanente biobank and found that policies limiting the return of research results pose significant barriers for participation [36]. As the current CHH/KS loci account for only about 50% of cases [6], further research is needed to fully chart the molecular basis of this rare condition. Additionally, patients may not appreciate that genetic testing for CHH/KS is not a singular, one-off experience. Cases may be explained years later when new CHH/KS loci are discovered, thereby enabling the identification of a causal variant explaining the case. The recent report from the National Academies of Sciences, Engineering, and Medicine provides a framework for returning research results to participants that may help to break down barriers to research participation and enhance research efforts [37]. Such models for engagement are needed to accelerate discovery and propel precision medicine for rare diseases such as CHH/KS. Notably, having to “chase down” results contributed to some participants feeling disrespected and unappreciated for their altruistic participation. Thus, there seems to be a disconnect between participant and researcher perspectives about perceived rights and responsibilities.

Finally, participants voiced ethical concerns regarding privacy and secondary use of return of results. Such elements are required to be described in informed consent documents. However, it is unclear whether consent documents clearly explicate this information in easy-to-understand lay terms. Indeed, evidence suggests that many participants do not understand informed consent for research studies and biobanks [38]. Thus, it appears that further work is needed to ensure comprehension of informed consent documents that support informed decisions. Further, if consent is treated as a legal document to be signed without discussion (rather than as a process), it is perhaps not surprising that participants felt unaware of the aspects of a study. Thus, a process must be followed in order to support decisions that are aligned with participant values and preferences. Employing user-centered design and principles of co-creation may offer ways to improve informed consent documents. Prior work has demonstrated the effectiveness of using co-creation with patients to develop high-quality patient-facing materials to help research participants better understand their research genetic test results [39].

A relative strength of the study is the sample size (*n* = 58), which is quite robust for qualitative research in rare-disease populations. However, there are several study limitations. A critique of qualitative inquiry relates to data saturation (i.e., the point at which no new codes emerge). To address this potential concern, we conducted six focus-group discussions. Typically, three to five focus-group discussions are needed to reach saturation [16]. Moreover, we sought to enhance the rigor of our approach by using external reviewers to validate the coding by independent investigators. An experienced researcher with expertise in qualitative methodology and the patient group leader reviewed the transcripts and coding and no new themes/sub-themes were identified. Traditionally, qualitative findings are not quantified. In the present study, we reported frequencies of themes/sub-themes to explore patient priorities (i.e., the most frequently arising codes) and inform the development of interventions to address the unmet patient needs. While not a traditional approach to qualitative inquiry, reporting frequencies in qualitative research has been employed to guide intervention development [31]. There is a possible bias of ascertainment in the study. It is possible that participants in patient meetings are those who seek more information about the condition and may not be representative of all CHH/KS. However, we have previously demonstrated that recruiting patients with CHH/KS in collaboration with patient support organizations does not recruit a monolithic group of patients with high needs [40]. It is possible that participants within a focus group can influence each other; therefore, themes/sub-themes could be more frequent within a particular focus group discussion. However, all five themes and 13/16 (81%) sub-themes appeared in each of the six focus group discussions. Thus, we are relatively confident in the fidelity of the focus group discussions. Few participants provided information on self-reported race/ethnicity. Thus, caution is merited in extrapolating findings to patients with CHH/KS in communities where other socio-cultural factors (i.e., race/ethnicity, social determinants of health, low/middle income countries) are key consideration for service design and uptake [41]. It is plausible that social determinants of health and the intersectionality (i.e., having a rare disease and belonging to a traditionally marginalized group) may affect patient and parent/guardian experiences with genetic testing [42].

## 5. Conclusions

Patients with CHH/KS are highly motivated by altruism in pursuing genetic testing—particularly in a research context. Many patients hope to shorten the ‘diagnostic odyssey’ and alleviate psychosocial effects of CHH/KS for future generations. Cost and providers who were unaware of genetic testing services were identified as significant barriers to genetic testing. The lack of pre-test decisional support/genetic counseling was identified as a significant unmet need as only 7% of participants had received genetic counseling. Focus group participants also expressed desire for post-test support to understand results and gain skills/confidence for initiating conversations with family members about CHH/KS. Notably, patient support groups often played an important role in patients finding information about genetic testing and for supporting intrafamilial communication about CHH/KS. Lastly, a more coherent approach to informed consent is needed to clarify ethical concerns, appropriately frame patient expectations, and delineate researcher responsibilities. Future directions may include web-based decisional support to increase access to genetic counseling, support high-quality decisions and promote intrafamilial communication regarding heritable conditions. Considering patient values, beliefs and experiences can inform more person-centered approaches to genetic testing for CHH/KS thereby fueling genetic discovery and propelling novel precision therapies for this rare condition.

## Figures and Tables

**Figure 1 jpm-12-00477-f001:**
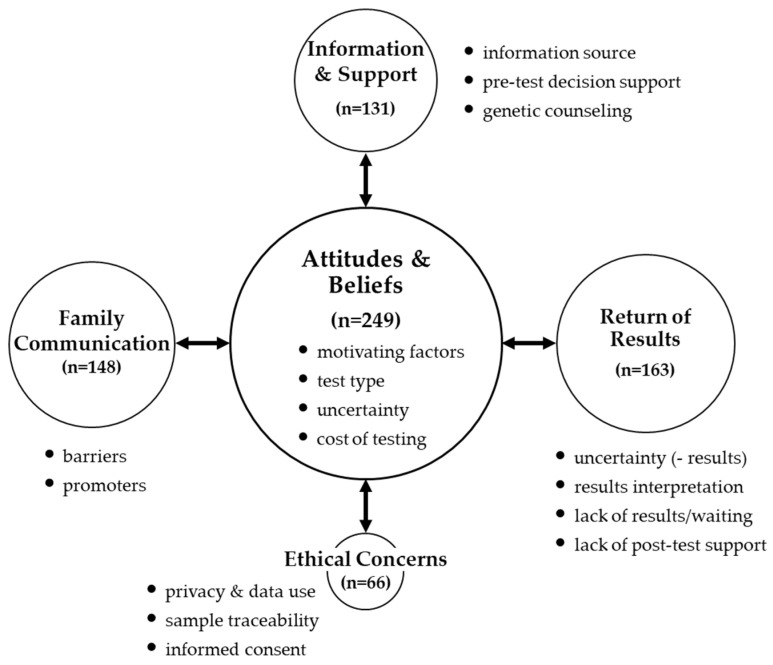
Themes and sub-themes of focus group discussions. The schematic depicts the major themes (circles) and sub-themes (bullets) across the six focus groups (*n* = 58). Circle size reflects the frequency of times the theme was noted in all groups. Pre-test experiences include the themes ‘information & support’ and ‘attitudes & beliefs.’ Post-test experiences include the themes ‘return of results’ and ‘family communication.’ The theme ‘ethical concerns’ (related to both pre- and post-test experiences) was the least frequently cited theme.

**Table 1 jpm-12-00477-t001:** Focus group participant characteristics (*n* = 58).

	In-Person (*n* = 33)	Virtual (*n* = 25)	Total (*n* = 58)
**male patients (*n*)**	19	11	30
age range (yrs.)	20–72	22–75	20–75
median age (yrs.)	37.0	60.0	40.5
mean ± SD	39.7 ± 14.4	53.4 ± 18.2 *	44.7 ± 17.0
**female patients (*n*)** **†**	6	12 *	18
age range (yrs.)	24–68	33–52	24–68
median age (yrs.)	33.0	43.0	40.0
mean ± SD	37.5 ± 16.2	42.8 ± 7.5	40.7 ± 11.5
**parents/guardians (*n*)**	8	2	10
age range (yrs.)	35–62	30–32	30–62
median age (yrs.)	47.5	31 *	45.0
mean ± SD	49.0 ± 9.9	-	45.4 ± 11.6

* *p* < 0.05 vs. in-person, † one virtual focus group was “female only” at the request of patients.

**Table 2 jpm-12-00477-t002:** Representative quotes relating to pre-test experiences: ‘attitudes & beliefs’.

Pre-Test Theme: ‘Attitudes & Beliefs’
**Sub-theme: ‘motivating factors’** Dimension: reasons for not testing “*I don’t think I’ve ever considered it […] I don’t think it would have changed anything about me and my condition and living with it.*” (male 4, virtual #1) “*I said, ‘well, when am I going to hear anything back about it?’ and when he [doctor] couldn’t answer any questions, that’s when I said, ‘well sorry’ [decided not to have testing].*” (female 1, in-person #1) Dimension: clearing up uncertainty “*When you get that result […] you think, ‘At least I know what the problem is so we can work on it’ […] I think it’s a relief, the diagnosis really. […] I think it’s a boost to, to be told what condition you have so you can deal with it.*” (male 3, virtual #1) Dimension: altruism (concern for family) “*I guess from a family perspective, it is incredibly important that we go ahead and get this genetic testing done… because, we then have that other decision to make… whether we go [on] having children ourselves.*” (male 5, virtual #3) “*Testing [is important] to help family advocacy… and information to trace the line [inheritance].*” (male 7, in-person #1) Dimension: altruism (concern for other patients) “*I don’t think there would be any one of us [in the meeting] that would want anyone else to have those experiences [late diagnosis of CHH/KS]. So, from my point of view […] anything that I can offer at all… 100% to progress medicine so that no one has to experience what I’ve been through in my lifetime.*” (male 7, virtual #3) “*Anything that can be done to help people... You know? Being diagnosed early… which likely will help with all these psychological issues. I think it’s… it’s worth it.*” (male 1, virtual #3) “*I would gladly submit samples for more genetic testing to further help treatments for future generations.*” (Male 5, virtual #1)
**Sub-theme: ‘uncertainty’** Dimension: uncertainty about results “*It’s going to come back [test results] and it’s [genetic variant] either going to be on my side […] linked to my side of the family… and I was going to have to feel guilty […] So, I kind of had mixed emotions.*” (mother 2, in-person #3) Dimension: incidental findings “*I think people who do genetic testing need to realize that there’s way different levels […] They don’t know there’s that secondary illness or comorbidity that could go along with it.*” (female 4, in-person #3)

**Table 3 jpm-12-00477-t003:** Representative quotes relating to pre-test experiences: ‘information & support’.

Pre-Test Theme: ‘Information & Support’
**Sub-theme: ‘information source’** Dimension: specialist (endocrinologist) “*Basically, the endocrinologist told us that it would be good to know.*” (mother 1, virtual #1) Dimension: online patient community *“… when I went on the groups on the Facebook [CHH/KS community]. That’s how I found out [about genetic testing].*” (male 3, virtual #2) Dimension: not offered testing “*We weren’t even offered genetic testing […] I would like to have been offered it […] I would have done it in a heartbeat.*” (female 2, virtual #3)
**Sub-themes: ‘decision support’/‘genetic counseling’** Dimension: lack of decisional support/genetic counseling “*At that time [diagnosis] there was no counselling… but even now, I’ve been offered none. I do feel it would be beneficial.*” “*Genetic counselling has never been offered to me.*” (female 4, virtual #2) Dimension: desired support/counseling “*I wish we could have gone somewhere where there was [genetic] counseling specific to this [CHH/KS] and [have] a workup as a family… […] but I think that’s almost impossible.*” (male 6, in-person #3) “*Counselling to deal with the information received… how to discuss [it] with family.*” (female 3, virtual #2)

**Table 4 jpm-12-00477-t004:** Representative quotes relating to post-test experiences: ‘return of results’.

Post-Test Theme: ‘Return of Results’
**Sub-theme: ‘uncertainty’** Dimension: not telling the “full story” “*Now that I know [the gene], I feel like I know a little bit… I got a piece of the puzzle but... still not the whole puzzle.*” (male 4, in-person #2) Dimension: negative test results “*Well, they can’t identify it […] what else would there be to say? […] Is there something else they should have told me?*” (male 7, virtual #1) Dimension: not knowing what to do with results “*knowing the gene that was mutated […] You’re going to find out this information and what can you do with it? […] maybe somebody in your family could know […] and it might help them. But what do you do with that information?*” (male 1, virtual #1)
**Sub-theme: ‘results interpretation’** Dimension: complexity of genetic information “*It [test results] may be clear to the practitioner […] whereas a layperson may completely misinterpret it and say that it’s an absolute, 100% chance… that it’s definitely going to happen.*” (male 3, in-person #1) Dimension: lack of interpretation support “*I think this is a role for genetic counselors who would come in and explain results […] Sending raw results to [patients] would be very scary. It [interpreting results] does need a genetic counselor I think to actually explain a lot of the stuff [results].*” (female 2, in-person #1)
**Sub-theme: ‘lack of results/waiting’** Dimension: extended waiting “*I had the genetic testing [at diagnosis] and just recently found out I have the PROKR2 mutation… after about 20 years.*” (female 2, virtual #2) “*I did [genetic testing] but it took years for… to get the results. […] that was really the only kind of frustrating part.*” (male 6, in-person #2) Dimension: lack of information/results “*In terms of information coming back… there wasn’t that much information that actually came back to me.*” (male 5, virtual #3) “*It would be just enough to make a difference to get the minimum information.*” (male 8, in-person #1) “*It seems like some doctors don’t think that their patients care about their certain condition… [it is] hard sending off blood work and not hearing anything back.*” (male 9, in-person 2) Dimension: chasing after results “*I gave blood [DNA] a few times. […] You even have to fight to find the results.*” (male 6, virtual #1) “*They [researchers] gave no definitive information back. I think that’s been the problem with all the studies. We’ve given our blood, we don’t get information back, and sometimes have to chase [researchers for results].*” (male 2, in-person #1) Dimension: feeling unvalued and disrespected “*[I was] sent off like a lab rat to have my bloods [drawn] and hear from no one until the next yearly review or until they want more blood for testing. It’s just [been] a very cold process for me.*” (female 1, virtual #3) “*We would like to emphasize [that] we would like to have the results. I feel like we’ve done our part.*” (male 2, in-person #3)
**Sub-theme: ‘lack of post-test support’** Dimension: lack of understanding about the impact of diagnosis “*They [providers] don’t understand because it’s so rare. We [patients] don’t know how to… you know, make them understand.*” (female 1, virtual #1) “*I think it’s hard to talk to other people about it because they can’t relate [to having CHH/KS].*” (mother 1, in-person #2) Dimension: desire for support in talking with family “*We need counselling to deal with the information received [test results]… how to discuss with family.*” (male 2, virtual #3) “*If there could have been a counselor that could have helped… because they [family] didn’t know what it [CHH/KS] was.*” (male 7, in-person #3) Dimension: guilt about not finding support “*It’s still hard to find counselors who are familiar with this [CHH/KS]…but we routinely beat ourselves up [because] we couldn’t find a counselor.*” (female 1, virtual #2)

**Table 5 jpm-12-00477-t005:** Representative quotes relating to post-test experiences: ‘family communication’.

Post-Test Theme: ‘Family Communication’
**Sub-theme: ‘barriers’** Dimension: unaccepting family members (e.g., denial, parental guilt) “*It’s a sexual problem and people don’t want to talk about it […] when you see your child is not developing... ‘oh, it can’t be’ […] ‘it can’t be my fault’ […] They don’t want to realize that, or don’t want to discuss it… so it becomes a secret.*” (male 3, virtual #3) “*You know, it’s like… everyone knew. I mean, no one talked about it… it was a big secret.*” (male 4, virtual #3) “*I just don’t want them [parents] to be upset and feel like, ‘because of me she was born like this’.*” (female 3, in-person #1) Dimension: Family dynamics (e.g., emotional distance, closed communication patterns) “*It’s so individual, and every family is different. So, there’s no easy way [to discuss CHH/KS].*” (male 9, in-person #1) “*I went through hell. […] over the years, it just it was a hard thing to deal with […] If the family’s not supportive, it’s going to be very hard for the person with Kallmann [syndrome].*” (male 3, virtual #3) “*I tried, I do try, [to tell family about CHH/KS]. I don’t know… I worry that they’re going to judge me.*” (male 6, virtual #1) Dimension: shame of having a rare disease “*I’ve never told my parents or my family, which I know is odd… But, it’s part of this […] shameful feeling.*” (male 1, virtual #3) “*Our [patients] whole life [sic] has been very private.. and it’s like we’re almost not supposed to talk about it.*” (female 4, virtual #2) “*Well, it’s a secret* with *me FROM my family. […] Yeah, I’ve never discussed it. Because… this is MY secret.*” (male 8, in-person #2)
**Sub-theme: ‘facilitators’ (promoters)** Dimension: family dynamics—open communication pattern “*It depends on your family. If you’re raised in a family which is pretty much open, you would naturally share with your family. But if you’re raised in a family that is very closed… it all depends on how you grow up and how you feel in your family. It’s very personal.*” (male 1, in-person #1) “*[what helped most with family] was their willingness to talk about it [CHH/KS]*”. (Mother 1, virtual #2) Dimension: altruism (concern for younger family members) “*I found it difficult to tell my family. I only told [them] in my mid-30′s when my cousin got engaged [to share possible risk].*” (male 4, in-person #1) “*I’ve only discussed it with my sister who was wanting to have kids. […] I told her because I just wanted her to have an informed decision. I don’t want to hide it from her, and then later, something goes on in one of her kids.*” (mother 1, in-person #2) Dimension: Patient peer support “*I Googled Kallmann syndrome and I discovered Neil [Patient leader] and that’s how I got in contact with the very first person that had Kallmann syndrome [sic]. Then, I was off to the meeting and that was my first experience with Kallmann syndrome. […] I think the main thing to remember is that we’re… we are MORE than Kallmann syndrome… We are people […] it [CHH/KS] doesn’t define you.*” (male 3, virtual 3) “*I think it’s such a level of secrecy because it’s so linked to sexuality and you don’t want to… you’re hiding it. You don’t want other people to know that you’re different. […] You don’t talk to your brothers about it either. I think that this group [patient support group] has helped a lot.*” (female 8, virtual #2) Dimension: Desired supports for communication (tools/resources) “*I’m just wondering if [technology] could be incredibly helpful and used as a tool to get family members on board [with genetic testing] I’m thinking […] you could arrange a meeting with someone, they could sort of loosely explain in layman’s terms, if you like, why this [genetic testing] is being done, how it’s being done.*” (male 5, virtual #3) “*I think what would encourage me [to talk to family] would be to have some sort of introductory video… and maybe some information books [or] leaflets. So, I can then… perhaps tell them about it and then show that introductory video about what the condition is… and maybe what the genetic test involves.*” (male 6, virtual #3)

**Table 6 jpm-12-00477-t006:** Representative quotes relating to the theme ‘ethical concerns’.

Theme: ‘Ethical Concerns’
**Sub-theme: ‘privacy & data use’** Dimension: privacy threats—research vs. for-profit “*I think it’s the difference between sacrificing your privacy for the sake of promoting science [research] and sacrificing your privacy for the sake of profit.*” (male 5, in-person #1) Dimension: data use “*Well, you know, once your gene sequence [is] in a database […] you kind of don’t know what’s going to… how that’s going to be used in the future.*” (male 4, virtual #3) Dimension: implications on health insurance “*…imagine if they sell that data to insurance providers or life insurance and then people would say, ‘your* insurance *[premium] has gone up’.*” (male 1, virtual #3)
**Sub-theme: ‘sample traceability & use’** Dimension: sample traceability “*Nobody knows where the blood goes after going to the doctor. I only tend to know a bit more, but the patients I talked to don’t know where their blood is going.*” (female 4, in-person #1) Dimension: sample use/secondary use “*It’s not just about being informed about what happened [sample use]… but it’s also BEFORE someone else uses your blood for testing. […] You can’t just take my sample and go off to the guy in another lab because he’s going to use the blood as well […] you have to ask me first if you can do that.*” (male 9, in-person #1)
**Sub-theme: ‘informed consent’** Dimension: receiving results “*You know, when you sign up for the tests they could ask at the beginning whether I want to know.*” (male 2, in-person #1) Dimension: secondary use “*You should have the ability the choose which further tests [are done].*” (female 1, in-person #2) Dimension: lack of consent process “*This is what my doctor asked, so… I just went in with a form and signed.*” (male 2, in-person #2)

## Data Availability

De-identified data will be made readily available upon request for research purposes to qualified individuals within the scientific community.

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
