# Peer review of "Exploring Rare Disease Patient Attitudes and Beliefs regarding Genetic Testing: Implications for Person-Centered Care"

_jpm, 2022, doi:10.3390/jpm12030477_

Round 1

Reviewer 1 Report

I find your manuscript very interesting. From a scientist's point of view, we mainly think in disease from a basic biology view, and we try to find the genetic causes to unveil the pathogenicity of the disease but often forget the patient's point of view, which is the most important. So, this paper is interesting to make a bridge among patients, scientists, and clinicians. 

I just have minor comments in the attached pdf.  

Reviewer 2 Report

This paper aims to describe patient experiences, attitudes and beliefs about genetic testing for congenital hypogonadotropic hypogonadism (CHH). The authors identified several themes that were related to pre- and post-test experiences and provide recommendations to improve person-centered care for CHH and other rare diseases. Overall, it is a very interesting and well-written manuscript. However, some minor points remain: 

  1. In table 1, page 4, it is shown that age and gender differ between in-person and virtual focus groups. For example, more females participated in virtual focus groups and mean age was higher for males who participated in virtual focus groups than for those who participated in in-person focus groups. Did the authors investigate the influence of age and gender on the themes, sub-themes and dimensions that emerged in the focus groups? Were there any differences? 
  2. There were two types of participants: patients with CHH and guardians/parents of children/adolescents with CHH. Did the authors notice any differences in themes, sub-themes and/or dimensions between the groups?
  3. Line 212, page 6: "Two dominant themes related to pre-test experiences". I think the authors mean post-test instead of pre-test.
  4. In the discussion, several limitations are presented (page 12). I can imagine that participants within a focus group can influence each other and that therefore themes/sub-themes/dimensions could be more frequent within that focus group. Could this be a limitation in this study or was there no difference between focus groups regarding the discussed themes/sub-themes/dimensions?
